# Associations between Traffic-Related Air Pollution and Cognitive Function in Australian Urban Settings: The Moderating Role of Diabetes Status

**DOI:** 10.3390/toxics10060289

**Published:** 2022-05-27

**Authors:** Rachel Tham, Amanda J. Wheeler, Alison Carver, David Dunstan, David Donaire-Gonzalez, Kaarin J. Anstey, Jonathan E. Shaw, Dianna J. Magliano, Erika Martino, Anthony Barnett, Ester Cerin

**Affiliations:** 1Mary MacKillop Institute for Health Research, Australian Catholic University, Melbourne, VIC 3000, Australia; amanda.wheeler@acu.edu.au (A.J.W.); alison.carver@acu.edu.au (A.C.); anthony.barnett@acu.edu.au (A.B.); 2Melbourne School of Population and Global Health, The University of Melbourne, Melbourne, VIC 3053, Australia; rachel.tham@unimelb.edu.au (R.T.); erika.l.martino@unimelb.edu.au (E.M.); 3Menzies Institute for Medical Research, College of Health and Medicine, University of Tasmania, Hobart, TAS 7000, Australia; 4School of Exercise and Nutrition Sciences, Deakin University, Burwood, VIC 3125, Australia; 5Baker Heart and Diabetes Institute, Melbourne, VIC 3004, Australia; david.dunstan@baker.edu.au (D.D.); jonathan.shaw@baker.edu.au (J.E.S.); dianna.magliano@baker.edu.au (D.J.M.); 6Barcelona Institute for Global Health, 08003 Barcelona, Spain; david.donaire@isglobal.org; 7School of Psychology, University of New South Wales, Randwick, NSW 2052, Australia; k.anstey@unsw.edu.au; 8Neuroscience Research Australia (NeuRA), Sydney, NSW 2031, Australia; 9School of Public Health and Preventive Medicine, Monash University, Melbourne, VIC 3004, Australia; 10School of Life Sciences, La Trobe University, Melbourne, VIC 3086, Australia; 11School of Public Health, The University of Hong Kong, 7 Sassoon Rd., Pokfulam, Hong Kong, China; 12Department of Community Medicine, UiT The Artic University of Norway, 9019 Tromsø, Norway

**Keywords:** air pollution, diabetes, cognitive function

## Abstract

Traffic-related air pollution (TRAP) is associated with lower cognitive function and diabetes in older adults, but little is known about whether diabetes status moderates the impact of TRAP on older adult cognitive function. We analysed cross-sectional data from 4141 adults who participated in the Australian Diabetes, Obesity and Lifestyle (AusDiab) study in 2011–2012. TRAP exposure was estimated using major and minor road density within multiple residential buffers. Cognitive function was assessed with validated psychometric scales, including: California Verbal Learning Test (memory) and Symbol–Digit Modalities Test (processing speed). Diabetes status was measured using oral glucose tolerance tests. We observed positive associations of some total road density measures with memory but not processing speed. Minor road density was not associated with cognitive function, while major road density showed positive associations with memory and processing speed among larger buffers. Within a 300 m buffer, the relationship between TRAP and memory tended to be positive in controls (β = 0.005; *p* = 0.062), but negative in people with diabetes (β = −0.013; *p* = 0.026) and negatively associated with processing speed in people with diabetes only (β = −0.047; *p* = 0.059). Increased TRAP exposure may be positively associated with cognitive function among urban-dwelling people, but this benefit may not extend to those with diabetes.

## 1. Introduction

The prevalence of cognitive decline, impaired cognitive function and neurocognitive diseases is increasing globally and represents a substantial burden of disease for individuals, families, communities and health systems [1]. Understanding the complex relationship of modifiable factors which contribute to cognitive decline is important so that impaired health can be avoided or delayed significantly. 

There is increasing evidence that air pollution is associated with lower cognitive function or cognitive decline [2], and in many countries traffic-related air pollution (TRAP) has substantially increased and is often the major source of ambient air pollution in residential areas [3]. Long-term exposure to TRAP can have diverse adverse effects on the central nervous system depending on the pollutant, dose, exposure period, health status and age of individuals [4]. Current mechanisms and physiological pathways linking TRAP to cognitive decline are not well-understood, but it has been suggested that exposure to fine particulate matter, diesel particles and gases such as ozone or nitrogen dioxide may enter the central nervous system via inhalation into the lungs and then travel through the blood circulatory system to the brain; or they can enter the brain directly by damaging and crossing the olfactory mucosa at the rear of the nose and directly into the brain [5,6]. On entering the central nervous system, these pollutants can trigger inflammation and the formation of free radicals that lead to oxidative stress, can damage nerve cells and promote the accumulation of tau and amyloid-B proteins in the brain, which can lead to neuronal dysfunction [7]. 

In addition to environmental factors, there are individual chronic conditions that can impact cognition. There is increasing evidence for middle-aged adults [8] and older adults [9] showing that poorly controlled type 2 diabetes mellitus (T2DM) increases the risk of late-life cognitive decline, Alzheimer’s disease (AD) and all-cause dementia [10,11]. Poorly controlled T2DM can lead to chronically elevated blood glucose levels which, in turn, are associated with vascular and neurodegeneration pathologies [12] through neuroinflammatory pathways [9]. 

Emerging evidence suggests that cardiovascular disease risk factors such as T2DM or hypertension may modify the relationship between exposure to air pollutants and neurological decline and neurodegenerative disease [13]. The nature of these relationships is not clear; however, it has been postulated that the effects that air pollutants have on impaired endothelial function, elevated systemic inflammation, mitochondrial dysfunction and oxidative stress [14] are also common to the effects of T2DM [15]. For example, a recent German study demonstrated a significant association between exposures to particulate air pollutants and elevated levels of glycated haemoglobin in red blood cells (HbA1c), which reflects the elevated mean plasma glucose levels over at least three months [16]. Chronically elevated blood glucose levels are associated with vascular and neurodegeneration pathologies [12]. While this suggests that adults with poor diabetic control are at greater risk for poorer cognitive function, it is not well-established if diabetes can modify the impact that air pollution exposure may have on cognitive function. With the prevalence of diabetes increasing globally [17], it is important to understand if having diabetes increases the adverse impacts of TRAP exposure on cognitive function. 

Findings from our previous study of the contribution of urban neighbourhood environments to cardiometabolic health and cognitive function [18] indicated that modelled ambient average nitrogen dioxide (NO_2_) levels (another proxy measure of TRAP) were directly positively associated with increased HbA1c levels (expβ = 1.003, *p* < 0.05) and with processing speed (cognitive function) (β = 0.243, *p* < 0.01). This finding with processing speed did not fit with the current understanding of the impact of TRAP on cognitive function [19,20] and raised the question of whether exposure misclassification could possibly explain these results. In the absence of objective measures of TRAP levels around these residential areas, to explore this relationship in greater depth, we revisited the data to establish if there were different proxy measures of TRAP that could be used to better understand the effects of this particular exposure on cognitive health for this population. 

The objectives of this study were two-fold: (1) to investigate whether TRAP, measured by neighbourhood road density at the residential address, is associated with cognitive function; and (2) if these effects differ between people with or without diabetes.

## 2. Materials and Methods

The details of the methods for this study have been previously reported [18]. In brief, this study used data from the Australian Diabetes, Obesity and Lifestyle (AusDiab) study, a three-wave national population-based survey designed to examine the prevalence, incidence and determinants of diabetes in Australian adults aged 25 years and over from 1999 to 2000 [21,22]. Participants were recruited using a stratified cluster sampling methodology from 42 randomly selected areas which consisted of 1296 contiguous census administrative units (Statistical Areas 1 [SA1]) across metropolitan and regional cities in seven Australian states/territories. SA1s have an average population size of 400 people. Eligible participants were aged ≥25 years, had lived at their address for at least 6 months, had no physical or intellectual disabilities and had provided written informed consent. This study used data from the second follow-up of AusDiab, conducted from 2011 to 2012, which was the only wave that collected data on cognitive function and for which relevant environmental exposures were available. As this study focused on urban environments, we excluded 473 participants who resided in towns or cities of <10,000 people from the analyses, which resulted in a final analytical sample of 4141 participants. 

This study was conducted according to the guidelines of the Declaration of Helsinki, and approved by the Alfred Hospital Ethics Committee, Melbourne, Australia: Ref. no 39/11; 2 March 2011. 

### 2.1. Measures

#### 2.1.1. Exposures: TRAP Measures

Proxy measures of exposure to TRAP sources were generated using Geographic Information Systems (GIS) software (ArcGIS v.10.5: ESRI, Redlands). We computed road density measures within Euclidean (aerial) polygon buffers of various sizes around geocoded participants’ residential addresses. Euclidean distances are calculated between two points, and the radii used to generate the buffers were: 200, 300, 500, 1000 and 1600 metres. These buffers were selected as they corresponded to the distance a mobile older person can walk within 30 min from home [23]. Road types were defined by PSMA Australia Ltd., Griffith, Australia [24] with major roads being 301 (national or state highway) or 302 (arterial road) or 303 (subarterial road); minor roads being 304 (collector road) or 305 (local road) or 306 (track). Road data were derived from Geoscape Australia’s 2012 Transport and Topography dataset [25]. The buffers included the density of all available roads, major roads only and minor roads only (m/km^2^). We also examined the straight-line distance, in metres, to the nearest ‘busy road’, which was defined as road types 301, 302 or 303. 

#### 2.1.2. Outcomes: Cognitive Function Measures

Processing speed and memory were the cognitive functions examined in this study as they are essential to reasoning and learning. Processing speed was measured using the Symbol–Digit Modalities Test (SDMT) [26], which required participants to use a reference key to find and report numbers corresponding to nine geometric figures as rapidly as possible in a 90 s period (score range: 0–60). Memory was assessed using the California Verbal Learning Test (CVLT) [27], in which participants recalled 16 common shopping items after a 20 min delay (score range: 0–16).

#### 2.1.3. Confounding Factors and Covariates

Causal directed acyclic graphs (DAGs) were used to inform the selection of a sufficient set of confounders to be included in the statistical analyses (Appendix A). The DAGs were based on previous research which established the hypothesised causal effects among the variables. The covariates included were age, sex, household income, area-level socio-economic status (SES), educational attainment, employment status, English-speaking background, living arrangements, neighbourhood self-selection indices for recreational purposes and accessibility to destinations/amenity, density of greenspace as calculated by the normalised difference vegetation index (NDVI), population density (numbers of persons per hectare) and a land use mix score related to outdoor physical activity that was computed to quantify the heterogeneity of residential, parkland and blue space (water bodies). Details of these measures are reported in Appendix A. Other self-reported variables which were considered in the causal DAG but not included in the final models because they were not identified as confounders included: history of cardiovascular disease, namely, angina, coronary heart disease or stroke; and tobacco smoking status (current smoker, previous smoker or never/non-smoker).

#### 2.1.4. Diabetes Status as an Effect Modifier

Diabetic status, as detected through oral glucose tolerance tests [28], was categorised as: diagnosed diabetes (physician-diagnosed diabetes mellitus and taking hypoglycemic medication or fasting plasma glucose (FPG) ≥ 7 mmol/L or 2 h plasma glucose (PG) ≥ 11.1 mmol/L); impaired glucose tolerance (IGT) (2 h PG ≥ 7.8 and < 11.1 mmol/L with FPG < 7.0 mmol/L); or impaired fasting glucose (IFG) (FPG ≥ 6.1 and < 7.0 mmol/L with 2 h PG < 7.8 mmol/L). Those with normal glucose tolerance were categorised as normal glucose tolerance controls (FPG < 6.1 mmol/L and 2 h PG < 7.8 mmol/L). 

### 2.2. Data Analytic Plan

Descriptive statistics were computed for all variables for the entire sample and by diabetes status. Over 21% of the cases had missing data on at least one variable, and missingness was related to individual-level socio-demographic characteristics, neighbourhood environmental characteristics and cognitive function (see Appendix A). As data were at least missing at random (MAR) rather than missing completely at random (MCAR) (see Appendix A), ten imputed datasets were created for the regression analyses as recommended by Rubin [29] and van Buuren [30]. Multiple imputations by chained equations were performed following currently recommended model-building and diagnostic procedures and using the package ‘mice’ [31] in R version 4.0.0 [32].

Associations of TRAP with cognitive function and the moderating effects of diabetes status on these associations were estimated using generalised additive mixed models (GAMMs) with random intercepts to account for possible curvilinearity in associations and clustering effects at the SA1 level [33]. As both cognitive function measures were approximately normally distributed, GAMMs used Gaussian variance and identity link functions. To test the moderating effects of diabetes status on TRAP–cognitive function associations, diabetes status (represented by two dummy variables with ‘normal glucose tolerance controls’ as the reference category) and its interaction terms with TRAP (two interaction terms, one for each dummy variable) were added to the main-effect models of TRAP and cognitive function. The overall statistical significance of the two interaction terms was determined using the F-test, which compared the fit of a GAMM with and without the two interaction terms [33]. Significant moderating effects of diabetes status (*p* < 0.05) were probed by estimating TRAP–cognitive function associations for the different categories of diabetes status (diabetes, IGT/IFG with normal glucose tolerance controls as the reference category). This was accomplished using appropriate linear combinations of regression coefficients derived from the relevant GAMMs [34] All regression analyses were performed on the imputed datasets (main analyses) as well as for only those participants with complete data (Appendix A).

## 3. Results

Participants with diagnosed diabetes or IGT/IFG were more likely to be of a non-English-speaking background, not employed, male, older, less educated, have a lower household income and reside in lower SES areas than normal glucose tolerance controls (Table 1). They also had lower scores on cognitive function tests and were more likely to have had heart problems or a stroke and be a current or ex-smoker. Participants diagnosed with diabetes tended to live in neighbourhoods with higher levels of TRAP, that is, higher road density within all buffers (Table 2) and dwelling density (Appendix A), than other participants, while normal glucose tolerance controls tended to reside in greener neighbourhoods, as defined using the NDVI (Appendix A).

Table 3 reports the adjusted associations of TRAP measures with cognitive function estimated from the total-effect models (using 10 multiple imputed datasets) described in Appendix A. We observed positive associations of total road density measures with memory but not processing speed. The associations were stronger for measures based on the smallest and two largest residential buffers. In general, minor road density was not significantly associated with cognitive function, while major road density showed positive associations with both memory and processing speed but only for measures based on larger residential buffers (Table 3). Distance to the nearest busy road was not significantly related to either cognitive function measures. Similar patterns of associations were observed in complete case analyses (Appendix A).

A few significant moderating effects of diabetes status on TRAP–cognition associations were observed, especially in the multiple imputation analyses (Table 4 and Appendix A). The F-ratio values reported in these tables represent tests of significance of the overall moderating effect of diabetes status on the relationships between TRAP measures and cognitive function. The moderating effects were represented by two regression coefficients: one estimating the difference in TRAP–cognition associations between normal glucose tolerance controls and participants with IGT/IFG and the other estimating the difference in TRAP–cognition associations between normal glucose tolerance controls and those with diabetes. The F-ratio tested the overall significance of both regression coefficients.

Interpretation of the multiple imputation analysis (Table 4) indicates that while the relationship between memory and major road density within 300 m residential buffers tended to be positive in normal glucose tolerance controls (β = 0.005; 95% CI: −0.0002, 0.011; *p* = 0.062), it was negative in those with diabetes (β = −0.013; 95% CI: −0.025, −0.002; *p* = 0.026). The same TRAP measure tended to be negatively related with processing speed in diabetics only (β = −0.047; 95% CI: −0.096, 0.002; *p* = 0.059). Major road density within 1600 m residential buffers was significantly positively associated with memory only in normal glucose tolerance controls (β = 0.019; 95% CI: 0.005, 0.033; *p* = 0.033) and unrelated to memory in those with IGT/IFG or diabetes (ps > 0.180).

## 4. Discussion

This confirmatory exploration of the impact of TRAP on cognitive function in urban Australian settings provided further mixed results with a range of proxy TRAP measures having positive or null associations with processing speed and memory amongst the full study sample. However, importantly, there were some indications that people with diabetes with higher major road density within 300 m of their residence, but not within other buffers, are at higher risk of poorer memory and processing speeds. Overall, however, most TRAP measures had null or positive associations with memory and processing speed amongst those with a diabetic status.

Multiple international studies examining air pollution and cognitive function have produced inconsistent results [35] and weak negative associations [20,36], but almost none have reported positive associations [19,37]. It is important to consider the high heterogeneity that exists between these studies with respect to the exposure assessment, the outcome assessments and the settings. Comparability or meta-analyses of findings have not been successfully conducted to date [19]. The absence of positive associations in the literature could be a consequence of publication bias as these findings do not support the biological evidence arising from animal studies, which indicates that there are plausible pathways through which air pollution may adversely impact cognitive function, as outlined in the Introduction.

The findings from this study indicate that those with a diabetic status of IGT/IFG or a diabetes diagnosis were at higher risk of having lower cognitive function if there was higher road density and associated traffic within 300 m of their residence, but not closer. To date, there are no published studies that have examined the impact of TRAP on cognitive function with diabetic status as a moderator, so comparison with other studies is not possible. Diabetes is associated with decrements in cognitive function [9] with higher levels of HbA1c (indicating medium- to long-term uncontrolled diabetes) associated with reduced processing speed and poorer memory [11]. These physiological factors alone may be impacting cognitive function, or their presence may be additive to the impact of TRAP on cognitive function. Disentangling the relationship between these factors will be important as TRAP is a modifiable risk factor but inequitably impacts those who are most disadvantaged.

The positive direct effects found in this study are inconsistent with the findings from our previous study, which used modelled NO_2_ as a proxy for exposure to TRAP and reported positive direct effects on processing speed but not memory [18]. These findings remain surprising given the evidence that currently exists that TRAP has weak or null associations with lower cognitive function [19].

Australian cities have relatively low levels of TRAP compared with most cities around the world [38]; however, air pollution levels are still above the standards recommended by the World Health Organization for protecting human health [39]. It is possible that if the air pollution levels were higher, the relationship between air pollution and these cognitive measures would be more negative, which would be in line with international findings.

We have hypothesised that these positive associations may exist due to higher road density being associated with easier access to cognition-enhancing destinations [18,40,41]. For example, commercial destinations that offer opportunities for cognition-enhancing activities, such as food outlets, shopping and entertainment venues, can be major sources of air pollution generated by high population volumes and, hence, traffic. Additionally, these streets may encourage walking and, although there is a risk of being exposed to more TRAP, the decision making required at street junctions may involve more cognitive processes [42] Additionally, street networks with greater connectivity have been associated with higher participation in walking for transport [43,44]. Further detailed examination of the built and natural environment in denser, destination-rich urban areas with more variability in TRAP would enable the differentiation of the positive effects of participating in cognition-enhancing activities from the potential negative effects of higher levels of TRAP.

### Strengths and Limitations

This study has utilised data from a significant Australian national cohort which has sampled from diverse urban environments in Australia. This cohort provided a unique opportunity to examine whether people with diabetes or impaired glucose tolerance were at risk of lower cognitive function if exposed to long-term TRAP. Although this wave of the AusDiab study was conducted 10 years ago, it is only now that environmental factors have now been retrospectively calculated and incorporated into the dataset so that complex relationships between select environmental exposures and cognitive function can be explored.

This study is limited by its cross-sectional design and the use of relatively coarse measures of TRAP exposure and land use characteristics. The TRAP exposures are proxy measures only and were not confirmed with objective measures of TRAP within the residential buffer zones. Hence, these exposure estimates may be subject to exposure misclassification. To overcome this, we used a national classification system for road types to harmonise the differences between different regions within Australia. This study did not include detailed information on access to cognition-enhancing destinations. As such, we are unable to disentangle impacts of the urban environment, which include positive effects on cognition associated with destinations which support healthy behaviours and negative effects, which include exposure to various pollutants or destinations which lead to unhealthy behaviours. This study does not include information on important factors such as genetic predisposition to cognitive impairment (for example, the presence of the ApoE4 lipoprotein gene [45]), or environmental exposures such as indoor air pollution [46] or exposure to noise [47].

Future research needs to address these limitations and examine the relationships in a longitudinal manner to identify the impacts of changes in exposures on cognitive function.

## 5. Conclusions

This study has demonstrated that, within Australian urban areas, increased levels of TRAP exposure proxies (road density) may be positively associated with cognitive function among urban-dwelling people, but this benefit may not extend to those with diabetes or impaired glucose metabolism. Further longitudinal and fine-grained research which incorporates measures of built and natural environmental exposures is needed to disentangle the effects of access to cognition-enhancing destinations and exposure to TRAP in those areas.

## Figures and Tables

**Table 1 toxics-10-00289-t001:** Descriptive statistics of the AusDiab sample 2011–2012.

Characteristic	Total Sample *n* = 4141	Diabetes Status		
		Diabetes (*n* = 405)	IGT/IFG (*n* = 620)	Normal Glucose Tolerance (*n* = 3003)
** *Socio-demographics* **				
Age (years), M ± SD	61.1 ± 11.4	67.2 ± 10.1	64.5 ± 11.4	59.6 ± 11.0
Sex, female, %	55.2	45.4	47.1	58.0
Educational attainment, %				
Up to secondary	32.7	39.3	41.1	30.1
Trade, associate diploma	43.6	41.0	31.13	44.2
Bachelor degree, postgraduate	23.1	18.3	17.1	25.2
Missing data	0.6	1.5	0.7	0.5
Employment status, %				
Not employed	30.4	47.7	38.9	26.4
Paid employment	52.2	34.8	45.2	56.3
Volunteer	15.1	14.6	13.4	15.7
Missing data	2.3	3.0	2.6	1.7
Household income (annual), %				
Up to AUD 49,999	32.9	47.9	39.0	29.9
AUD 50,000–AUD 99,999	26.8	21.5	26.0	28.0
AUD 100,000 and over	28.9	14.6	23.1	32.1
Missing data	11.5	16.0	11.9	10.0
Living arrangements, %				
Couple without children	48.2	50.9	53.6	47.2
Couple with children	26.8	15.6	21.1	29.4
Other	22.4	30.4	22.6	21.6
Missing data	2.4	3.2	2.7	1.8
English-speaking background, %	89.9	83.7	87.4	91.5
Area-level IRSAD, M ± SD	6.4 ± 2.7	5.9 ± 2.8	6.3 ± 2.7	6.5 ± 2.7
Residential self-selection—access to destinations, M ± SD	3.0 ± 1.4	3.0 ± 1.3	3.0 ± 1.3	2.9 ± 1.3
Missing data, %	9.0	9.4	10.5	8.0
Residential self-selection—recreational facilities, M ± SD	3.1 ± 1.5	3.0 ± 1.6	3.1 ± 1.5	3.1 ± 1.4
Missing data, %	8.7	9.1	10.2	7.7
** *Health-related variables* **				
Diabetes status, %				
Diabetes	9.8	-	-	-
IGT/IFG	15.0	-	-	-
Normal glucose tolerance	72.5	-	-	-
Missing data, %	2.7	-	-	-
Heart problems/stroke history, %	8.7	19.8	10.5	6.8
Missing data, %	1.0	1.5	0.2	0.0
Tobacco-smoking status, %				
Current smoker	7.0	4.2	8.7	7.2
Previous smoker	35.9	44.0	37.6	34.4
Non-smoker	54.5	48.6	51.0	56.6
Missing data, %	2.6	3.2	2.7	1.8
***Cognitive function***, M ± SD				
Memory, CVLT score	6.5 ± 2.4	5.6 ± 2.4	6.2 ± 2.4	6.7 ± 2.4
Missing data, %	2.3	5.2	1.5	1.7
Processing speed, SDMT score	49.7 ± 11.6	43.6 ± 12.4	47.2 ± 12.1	51.1 ± 11.0
Missing data, %	2.0	4.0	1.3	1.6

M, mean; SD, standard deviation; IRSAD, Index of Relative Socioeconomic Advantage and Disadvantage; IGT, impaired glucose tolerance; IFG, impaired fasting glucose; CVLT, California Verbal Learning Test; SDMT, Symbol–Digit Modalities Test.

**Table 2 toxics-10-00289-t002:** Descriptive statistics of traffic-related air-pollution measures (M ± SD).

Characteristic	Total Sample *n* = 4141	Diabetes Status		
		Diabetes (*n* = 405)	IGT/IFG (*n* = 620)	Normal Glucose Tolerance (*n* = 3003)
Road density (100 m/km^2^)				
200 m Euclidean buffer	117.1 ± 42.0	120.7 ± 42.9	116.8 ± 42.4	117.1 ± 41.8
300 m Euclidean buffer	114.9 ± 40.1	118.9 ± 39.0	114.6 ± 39.7	114.6 ± 40.4
500 m Euclidean buffer	107.4 ± 37.7	111.0 ± 35.5	107.4 ± 36.5	107.1 ± 38.2
1000 m Euclidean buffer	95.8 ± 34.0	99.5 ± 33.2	96.6 ± 33.2	95.1 ± 34.2
1600 m Euclidean buffer	87.6 ± 32.3	91.2 ± 31.9	89.1 ± 31.5	86.8 ± 32.4
Minor road density (100 m/km^2^)				
200 m Euclidean buffer	88.6 ± 36.7	92.1 ± 37.6	89.3 ± 36.4	88.1 ± 36.7
300 m Euclidean buffer	83.0 ± 32.7	86.7 ± 32.4	84.0 ± 32.9	82.4 ± 32.6
500 m Euclidean buffer	75.7 ± 29.1	79.3 ± 28.8	77.1 ± 28.9	75.0 ± 29.1
1000 m Euclidean buffer	66.1 ± 26.3	70.0 ± 26.3	67.9 ± 25.8	65.1 ± 26.2
1600 m Euclidean buffer	59.6 ± 25.2	63.6 ± 25.3	61.7 ± 24.9	58.5 ± 25.2
Major road density (100 m/km^2^)				
200 m Euclidean buffer	9.7 ± 16.9	10.6 ± 17.7	9.6 ± 17.5	9.5 ± 16.6
300 m Euclidean buffer	13.3 ± 17.0	14.7 ± 17.2	13.7 ± 17.7	13.1 ± 16.9
500 m Euclidean buffer	13.6 ± 13.3	14.6 ± 13.4	13.9 ± 13.4	13.4 ± 13.3
1000 m Euclidean buffer	13.1 ± 8.9	14.0 ± 9.0	13.4 ± 9.0	13.0 ± 8.9
1600 m Euclidean buffer	12.4 ± 7.4	12.9 ± 7.5	12.8 ± 7.6	12.2 ± 7.3
Distance to nearest busy road (100 m)	4.57 ± 4.98	4.38 ± 5.05	4.48 ± 4.86	4.59 ± 4.91
NO_2_ (ppb)	5.53 ± 2.05	5.68 ± 2.10	5.47 ± 1.88	5.50 ± 2.07

M, mean; SD, standard deviation; IGT, impaired glucose tolerance; IFG, impaired fasting glucose; ppb, parts per billion.

**Table 3 toxics-10-00289-t003:** Associations of transport-related air pollution (TRAP) measures with cognitive function (multiple imputation analyses; all participants *n* = 4141).

TRAP Measures	Memory (CVLT Score)			Processing Speed (SDMT Score)		
	*β*	95% CI	*p*	*β*	95% CI	*p*
Road density (100 m/km^2^)						
200 m Euclidean buffer	**0.003**	**0.001, 0.005**	**0.008**	0.005	−0.002, 0.012	0.173
300 m Euclidean buffer	** *0.002* **	** *−0.0003, 0.004* **	** *0.088* **	0.005	−0.003, 0.013	0.257
500 m Euclidean buffer	** *0.002* **	** *−0.0003, 0.004* **	** *0.084* **	0.003	−0.006, 0.012	0.507
1000 m Euclidean buffer	**0.003**	**0.001, 0.006**	**0.018**	0.002	−0.008, 0.013	0.678
1600 m Euclidean buffer	**0.004**	**0.001, 0.007**	**0.015**	0.007	−0.005, 0.018	0.263
Minor road density (100 m/km^2^)						
200 m Euclidean buffer	** *0.002* **	** *−0.0001, 0.004* **	** *0.058* **	0.001	−0.008, 0.010	0.831
300 m Euclidean buffer	0.001	−0.002, 0.004	0.424	−0.0004	−0.010, 0.010	0.932
500 m Euclidean buffer	0.001	−0.002, 0.004	0.686	−0.009	−0.021, 0.003	0.145
1000 m Euclidean buffer	0.002	−0.002, 0.005	0.425	−0.012	−0.026, 0.003	0.106
1600 m Euclidean buffer	0.001	−0.003, 0.006	0.510	−0.008	−0.024, 0.008	0.311
Major road density (100 m/km^2^)						
200 m Euclidean buffer	0.003	−0.002, 0.007	0.268	0.005	−0.012, 0.023	0.558
300 m Euclidean buffer	0.001	−0.003, 0.006	0.614	0.006	−0.012, 0.024	0.535
500 m Euclidean buffer	** *0.005* **	**−*0.001, 0.011***	** *0.097* **	0.020	−0.005, 0.044	0.116
1000 m Euclidean buffer	**0.010**	**0.001, 0.020**	**0.038**	0.034	−0.007, 0.079	0.101
1600 m Euclidean buffer	**0.014**	**0.002, 0.026**	**0.026**	**0.055**	**0.004, 0.105**	**0.036**
Distance to nearest busy road (100 m)	−0.0004	−0.017, 0.016	0.963	−0.054	−0.118, 0.011	0.102

β, regression coefficient; CI, confidence intervals; *p*, *p*-value; CVLT, California Verbal Learning Test; SDMT, Symbol Digit Modality Test. Estimates of regression coefficient adjusted for covariates listed in Appendix A. In bold are statistically significant associations at a probability level of 0.05. In bold italics are statistically significant associations at a probability level of 0.10.

**Table 4 toxics-10-00289-t004:** Moderation effects of diabetes status on the associations between traffic-related air pollution measures with cognitive function (multiple imputation analyses; *n* = 4141).

TRAP Measures	Memory (CVLT Score)		Processing Speed (SDMT Score)	
	*F* (2, 4114)	*p*	*F* (2, 4114)	*p*
Road density (100 m/km^2^)				
200 m Euclidean buffer	0.75	0.472	0.27	0.766
300 m Euclidean buffer	0.96	0.383	0.57	0.566
500 m Euclidean buffer	0.56	0.573	0.61	0.542
1000 m Euclidean buffer	1.18	0.307	1.93	0.145
1600 m Euclidean buffer	1.51	0.221	** *2.60* **	** *0.074* **
Minor road density (100 m/km^2^)				
200 m Euclidean buffer	0.22	0.800	0.30	0.741
300 m Euclidean buffer	0.20	0.818	0.28	0.753
500 m Euclidean buffer	0.36	0.701	0.20	0.819
1000 m Euclidean buffer	0.20	0.815	1.46	0.231
1600 m Euclidean buffer	0.08	0.921	** *2.94* **	** *0.053* **
Major road density (100 m/km^2^)				
200 m Euclidean buffer	2.22	0.108	0.61	0.543
300 m Euclidean buffer	**4.80**	**0.008**	**2.98**	**0.050**
500 m Euclidean buffer	2.28	0.102	1.16	0.314
1000 m Euclidean buffer	1.76	0.172	0.57	0.567
1600 m Euclidean buffer	**3.16**	**0.043**	1.66	0.190
Distance to nearest busy road (100 m)	0.84	0.432	1.33	0.265

*Notes.* F, F-ratio; *p*, *p*-value; CVLT, California Verbal Learning Test; SDMT, Symbol Digit Modality Test. Estimates of regression coefficient (β) adjusted for covariates listed in Appendix A. In bold are statistically significant associations at a probability level of 0.05. In bold italics are statistically significant moderation effects of diabetes status on TRAP–cognitive function at a probability level of 0.10.

## Data Availability

Data that support the findings of this study are available on request under a license agreement. Written applications can be made to the AusDiab Steering Committee (Dianna.Magliano@baker.edu.au).

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
