# Peer review of "Associations between Traffic-Related Air Pollution and Cognitive Function in Australian Urban Settings: The Moderating Role of Diabetes Status"

_toxics, 2022, doi:10.3390/toxics10060289_

Round 1

Reviewer 1 Report

Thank you for the opportunity to review this manuscript. The research is well designed.

Since the database reports on results gathered a decade ago, additional justification as to why this was appropriate would help improve the context for the readers. There is also scope for a more rallying call to include more measures pertinent to planetary health in future iterations of AusDiab.

Reviewer 2 Report

I'm troubled by the finding that although lower cognition was related to residence within 300 meters of elevated road traffic, it was not associated with distance from from the source. This finding is inconsistent with the association of asthma in children residing proximate to roads with heavy traffic and directly related to proximity to the sources.

The authors recognize that mechanistically, inflammation and oxidative stress apply to cognition decline. Both inflammation and OS are dose related, so one would expect distance related effects in the current study. The inconsistency noted needs to be further addressed prior to publication.

Reviewer 3 Report

The paper describes the associations between traffic-related air pollution and cognitive function in Australian urban settings: the moderating role of diabetes status. This study suggests that increased TRAP exposure may be positively associated with cognitive function among urban-dwelling people, but this benefit may not extend to those with diabetes.

However, the article lacks scientific validity, and accuracy of explanation of research method and research result.

  1. In line 100-110, sample selection is not clear. Is it a random selection? “Third wave of AusDiab” is also not clear. References of no. 21 and 22 stated that baseline survey in 1999-2000, the first follow-up in 2004-2005 and the second follow-up in 2011-2012. Was the third wave the third follow-up? Author should clarify.

  1. If my understanding is correct, "in 1999-2000" should be added just after "aged 25 years and over" in line 104.

  1. In Figure S1, the abbreviation is used, but readers don’t understand the meaning of the abbreviation. The author should explain the abbreviation.

  1. In line 138-148, and in the Section S1 Detailed description of covariates, and table 1, the explanations and classifications of covariates do not match.

  1. Table 1 shows Heart problem, Tabaco-smoking status, but there is no explanation in the Methods section.

  1. The heading "effect modifiers" in 2.1.4 should be “Diabetes (or Diabetes) status.

  1. In line 171, what the meaning of “SA1 level”. Please clarify.

  1. In table 2, the meaning of 200m / 300m / 500m / 100m / 1600m Eulidean buffer is unknown. Does "1600m Eulidean buffer" mean "road length within a radius of 1600m from the subject's home"? The explanation is insufficient.

  1. In line 208, table 3 footnote, author mentioned “Estimates of regression coefficient adjusted for covariates listed in Table S2.” Is Table S2 a mistake in Table S1?

  1. The title and purpose of the paper is "moderate role of diabetes status", and Table S5 showing the result should be included in the text, not in Supplemental. In addition, the explanation of the analysis method of moderate is insufficient in Chapter 2.2.
    Table S1 explains TE (total-effect models) and MD (models examining the moderating effects of diabetes), but the reader does not know which table each analysis results is.
    Also, if my understanding is correct, Table S1 states that diabetes status is included in the covariates in the MD model, while β shows the difference in outcome scores between those without diabetes and those with diabetes, doesn’t it? Does β shows moderate effects of diabetes status?

  1. Risk factors for outcome cognitive function should be considered as covariates in the analysis, for instance smoking status, BMI and so on.

Round 2

Reviewer 3 Report

I confirmed the authors' corrections.